# GRADIENT REGULARIZATION-BASED CROSS-PROMPT ATTACKS ON VISION LANGUAGE MODELS

## ABSTRACT

Recent large vision language models (VLMs) have gained significant attention for their superior performance in various visual understanding tasks using textual instructions, also known as prompts. However, existing research shows that VLMs are vulnerable to adversarial examples, where imperceptible perturbations added to images can lead to malicious outputs, posing security risks during deployment. Unlike single-modal models, VLMs process both images and text simultaneously, making the creation of visual adversarial examples dependent on specific prompts. Consequently, the same adversarial example may become ineffective when different prompts are used, which is common as users often input diverse prompts. Our experiments reveal severe non-stationarity when directly optimizing adversarial example generation using multiple prompts, resulting in examples specific to a single prompt with poor transferability. To address this issue, we propose the Gradient Regularized-based Cross-Prompt Attack (GrCPA), which leverages gradient regularization to generate more robust adversarial attacks, thereby improving the assessment of model robustness. By exploiting the structural characteristics of the Transformer, GrCPA reduces the variance of back-propagated gradients in the Attention and MLP components, utilizing regularized gradients to produce more effective adversarial examples. Extensive experiments on models such as Flamingo, BLIP-2, LLaVA and InstructBLIP demonstrate the effectiveness of GrCPA in enhancing the transferability of adversarial attacks across different prompts.

## 1 INTRODUCTION

Large Vision Language Models (VLMs), such as GPT-4 (OpenAI, 2023b), have recently garnered substantial interest from the AI research community. Unlike Large Language Models (LLMs), which are limited to processing plain text (OpenAI, 2023a), VLMs can interpret image inputs and perform a range of visual understanding tasks guided by textual instructions, or prompts. These tasks include image captioning (Li et al., 2023a; Zhang et al., 2020; Sheng et al., 2021), information extraction (Liu et al., 2024; Li et al., 2023b), complex counting (Bavishi et al., 2023), and visual grounding (Wang et al., 2023a; Bai et al., 2023), among others. This powerful multimodal perception capability has facilitated the deployment of more models in real-world production environments.

Recent studies have revealed that VLMs are susceptible to attacks from adversarial examples (Gu et al., 2023; Madry et al., 2018; Szegedy et al., 2014; Li et al., 2024a; Mahmood et al., 2021; Mao et al., 2023; Yu et al., 2023; Wang et al., 2023b; Shayegani et al., 2023). These attacks involve the addition of imperceptible disturbances to clean images, which can induce the models to output malicious content. Such adversarial attacks can circumvent the security constraints of LLMs or even embed advertising information into images (Niu et al., 2024; Qi et al., 2023; Bailey et al., 2023; Lu et al., 2024; Yuan et al., 2023). Therefore, designing effective attack methods to identify potential vulnerabilities before deploying VLMs in security-related applications is of paramount importance (Li et al., 2024b; Gao et al., 2024b; Wang et al., 2023c).

Adversarial attacks can be broadly categorized into white-box attacks and black-box attacks (Gao et al., 2024a; Cheng et al., 2019). A white-box attack refers to an attacker who has access to all the structural and weight information of the model (Ebrahimi et al., 2018). Conversely, a black-box attack refers to an attacker who can only access the model's external usage interface (Guo et al.,

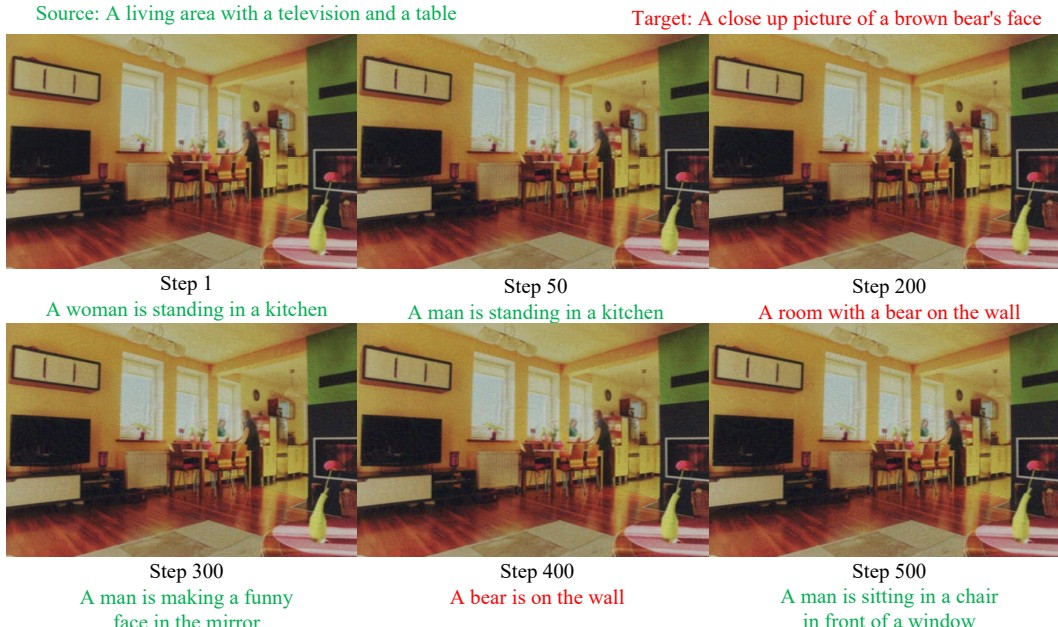

Figure 1: Illustration of the attack iteration process. Green represents the clean image description, while red represents the attack target. Adversarial attacks are unstable during the iteration process, causing fluctuations around the attack target.

2019). Given their expanded operational scope and potential to transition into black-box attacks, white-box attacks are consequently receiving heightened attention (Weng et al., 2024).

In contrast to widely studied classification models, adversarial attacks on VLMs present a more complex challenge. These attacks can stem from two perspectives: visual input and textual input. However, visual attacks are often imperceptible to users and occupy a continuum of disturbance space, making them more commonly utilized. The generation of visual adversarial examples for VLMs requires coordination with specific prompts. In other words, the same visual adversarial sample may not be effective when encountering diverse prompts (Cui et al., 2023). This phenomenon is prevalent during the model deployment phase, as users tend to input prompts based on their individual language preferences. Therefore, this paper focuses on cross-prompt visual adversarial attacks.

An intuitive method to enhance across-prompts transferability is to utilize multiple prompts in the iterative generation of adversarial examples (Moosavi-Dezfooli et al., 2017). However, in our experiments, we identified three issues with this approach: (a) A serious non-stationary phenomenon is observed, characterized by large fluctuations in the success rate of the attack during the iteration of adversarial samples, as shown in Figure 1. We attribute this to overfitting during optimization, since adversarial attacks on VLMs usually require a large number of iterations, such as 10,000, to succeed, causing adversarial examples to become specific to their conditions (model and prompt) (Schlarmann & Hein, 2023). (b) The calculation of text loss is extremely sensitive. Initially, we inadvertently computed the loss for the entire sequence using teacher forcing, but found the results largely unsuccessful. Subsequently, we recognized the need to focus solely on the loss pertaining to the model's output section. (c) Methods based on image classification for enhancing transferability are not adaptable to VLMs. We tested methods like MI-FGSM (Dong et al., 2018), Input Diversity (Xie et al., 2019), Variance Tuning (Wang & He, 2021), and found that the transferability across prompts did not increase, but even decreased. A more detailed analysis can be seen in the Appendix A.1.

Based on these observations, we contend that the design of visual adversarial examples for VLMs should take into account both image and text inputs comprehensively, with a particular focus on mitigating overfitting in the textual domain. VLMs often integrate substantial language models, which consist of numerous Transformer blocks, potentially leading to learned features that are specific to the prompts or the model itself (Wang & He, 2021). In this paper, we introduce a **G**radient **R**egularization-based **C**ross-**P**rompt Attack (**GrCPA**) method designed to alleviate overfitting of both

visual and textual features within the LLM' Transformer blocks, thereby enhancing the transferability of visual adversarial examples. Specifically, we implement gradient clipping on both visual and textual features during the loss back-propagation phase to counter overfitting. Note that we modify only a very small number of gradients, which does not affect the overall convergence of the chain rule (Zhang et al., 2023a; Wei et al., 2022).

To verify the effectiveness of GrCPA, we employ prompts from three distinct vision-language tasks: image classification, image captioning, and visual question answering (VQA). We evaluate our method's efficacy on well-known VLMs, including Flamingo (Alayrac et al., 2022), BLIP-2 (Li et al., 2023a), LLaVA-1.5 (Liu et al., 2023) and InstructBLIP (Dai et al., 2023). The experimental results indicate that GrCPA exhibits superior attack performance and enhanced transferability.

Overall, the main contributions of this paper include:

1. To the best of our knowledge, we first identify the non-stationary phenomenon in adversarial attacks against vision language models, and argue that its essence is overfitting in the optimization process. We also attempt previous enhancement methods for single-modal models and find them to be ineffective.

2. We propose a gradient regularization method to enhance the transferability of visual adversarial examples, thus effectively alleviating overfitting issues in the deep Transformer blocks of visual and textual features.

3. We validate the effectiveness of our method through detailed experiments and provide a new perspective for future attacks against VLMs.

## 2 RELATED WORK

**Adversarial Transferability.** Szegedy et al. (2014) first proposed the concept of adversarial examples, revealing the vulnerability of neural networks. The transferable attacks, which have widespread impacts in the real world, have triggered a large number of subsequent studies Cheng et al. (2020); Wu et al. (2022); Zhang et al. (2023b); Chakraborty et al. (2021); Madry et al. (2018); Xu et al. (2022). Previous work has primarily focused on classification models, with an emphasis on enhancing transferability through gradient optimization, input augmentation. Gradient optimization methods, led by the Fast Gradient Sign Method (FGSM) (Goodfellow et al., 2015), along with its derivatives such as Iterative FGSM (I-FGSM) (Kurakin et al., 2017), Projected Gradient Descent (PGD) (Madry et al., 2018), Momentum Iterative FGSM (MI-FGSM) (Dong et al., 2018), among others, have emerged as prominent techniques in the literature. On another front, input augmentation primarily involves applying various transformations to the input image at each iteration, such as random resizing and padding, as seen in methods like DIM Xie et al. (2019), SIM Lin et al. (2020), and TIM Dong et al. (2019). We endeavor to improve the transferability of attacks on VLMs utilizing traditional methods, but observe no substantial enhancement in their effectiveness. This highlights the complexity and inherent challenges of multimodal tasks, prompting a reevaluation of our previous research methodologies.

**Adversarial Robustness of Vision Language Models.** Alongside the proliferation of large VLMs, the associated security research has garnered significant attention Gao et al. (2024a); Sun et al. (2024); Ni et al. (2024); Zhang et al. (2024); Guo et al. (2024); Luo et al. (2024b); Zhou et al. (2024); Cheng et al. (2024); Wang et al. (2024). For example, Zhao et al. (2023) induce misinterpretation of image content in models such as BLIP-2 through black-box attacks. There is also a body of work utilizing adversarial attacks to circumvent security alignment of LLM components (Bagdasaryan et al., 2023; Carlini et al., 2023; Niu et al., 2024; Qi et al., 2024), posing security risks to VLMs. The work closest to ours is CroPA Luo et al. (2024a), which turns the generation of visual adversarial examples into a max-min process, achieving significant improvements. Our method from the perspective of reverse gradient is orthogonal to it, with more flexible and simpler operational methods.

## 3 METHOD

In this section, we first introduce adversarial attack setup against VLMs. Then, we formally present baseline methods for generating visual adversarial examples using a single prompt and multiple

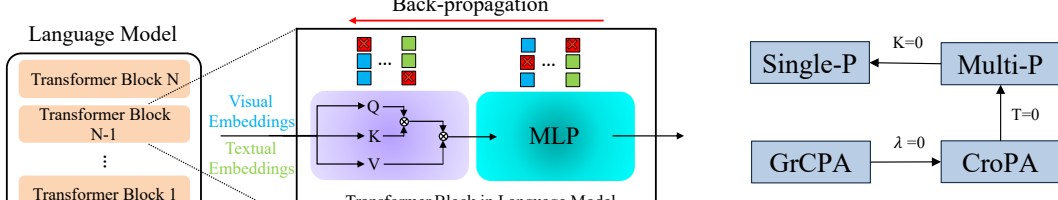

(a) Illustration of gradient regularization.

(b) Relationships among various methods.

Figure 2: Overview of our proposed method. (a) The process of gradient regularization involves performing clipping during back-propagation in the Transformer blocks of the LLM for both visual embeddings and text embeddings, specifically by attenuating the extreme values of gradients in each token. (b) By adjusting different parameter values, various methods can be transformed.

prompts. Finally, we introduce our proposed GrCPA method, which enhances cross-prompt transferability through gradient regularization during backpropagation.

## 3.1 THREAT MODEL

Without loss of generality, VLMs complete a series of downstream tasks through visual question answering (VQA). The model is provided with an image $v$ and question $q$ (i.e., prompts), and in response, it generates an answer $a$. We denote a VLM with the function $f_\theta$, where $\theta$ represents its parameters, such that $a = f_\theta(v, q)$.

Thus, the threat model for adversarial attacks on vision language models can be represented as:

**Adversarial knowledge** refers to the information an attacker has about the target model. In this paper, we focus on white-box attacks, where we have full access to all details of the victim's model, including its architecture and weights. This access allows us to leverage the gradients obtained through backpropagation to generate adversarial examples effectively.

**Adversarial goals** describe the malicious objectives that an attacker aims to achieve, typically categorized into targeted and untargeted attacks. For VLMs, targeted attacks seek to induce the model to output specific content, including bypassing alignment constraints. In contrast, untargeted attacks aim to provoke the model into producing incongruous responses. Since untargeted attacks can often be achieved through targeted attacks, this paper places greater emphasis on targeted attacks.

**Adversary capabilities** refer to the resources and constraints available to the attacker. To ensure that adversarial examples remain imperceptible to humans, the image perturbation $\delta$ is constrained by $||\delta||_p \le \epsilon$, where $\epsilon$ represents the magnitude of the perturbation and $||\cdot||_p$ denotes the $L_p$ norm. This paper primarily employs the $L_\infty$ norm to align with previous work (Luo et al., 2024a).

## 3.2 BASELINE METHODS

To induce the model to output specific content, given a query $q$ and a target answer $a$, we optimize the loss of the language model with respect to the $(q, a)$ pair and backpropagate it to the image, thus generating adversarial examples. We denote this method as **Single-P**. However, the activation of adversarial examples generated by this method also depends on the optimization of $q$ used during the process. In other words, if replaced with another prompt $q'$, the model may fail to produce $a$ in response (Cui et al., 2023).

To enhance the cross-prompt transferability of visual adversarial samples, a straightforward approach is to use multiple prompts during optimization (Moosavi-Dezfooli et al., 2017). Given a set of textual prompts $\mathcal{Q} = \{q_1, q_2, q_3, \ldots, q_M\}$, we induce the model to output the predetermined target answer $a$ under the query of each item in $\mathcal{Q}$ in the presence of adversarial noise $\delta$. Specifically, we minimize the following language modeling loss:

$$\min_{\delta_v} \sum_{i=1}^{K} \mathcal{L}(f(v + \delta, q_i), a) \tag{1}$$

Where $\mathcal{L}$ is typically the cross-entropy loss. Note that we compute the loss for only the part corresponding to answer $a$ rather than the entire $(q, a)$ sequence. We refer to this method as **Multi-P**.

It is evident that the improvement in cross-prompt transferability is directly proportional to the increase in the number of textual prompts, denoted by $K$. However, exhaustively exploring all potential prompts is often impractical due to the significantly increased computational complexity. Therefore, it is essential to enhance transferability with a limited number of prompts. To address this, **CroPA** Luo et al. (2024a) proposes using a set of learnable prompts to update the visual adversarial perturbation, aiming to counteract the misleading effects of adversarial images:

$$\min_{\delta_v} \max_{\delta_t} \mathcal{L}(f(v + \delta_v, q_i + \delta_t), a) \tag{2}$$

where $\delta_v$ represents the perturbation added to the image, while $\delta_t$ represents the perturbation added to the text. To smooth the optimization process, a text perturbation update frequency $T$ is introduced. This means that for every $T$ updates of the visual perturbation, the text perturbation is updated once.

### 3.3 GRCPA

Orthogonal to CroPA, we introduce GrCPA, which focuses on gradient regularization during the backpropagation process. Visual adversarial attacks fundamentally involve optimizing images using gradient descent. Consequently, large gradients can lead to local optima and trigger overfitting issues (Wang & He, 2021). This motivates us to clip the gradients of both visual and text features, thereby enhancing cross-prompt transferability.

Existing large VLMs typically consist of three components: a visual encoder, a projection layer, and an LLM. The image passes through the visual encoder to obtain a set of features, which are then aligned to the input space of the LLM by the projection layer to form visual tokens. These visual tokens are concatenated with textual tokens and fed into the LLM for autoregressive generation. The LLM is composed of multiple stacked Transformer blocks, with each block consisting of Attention and MLP components (Vaswani et al., 2017).

Given the gradient vector $G \in \mathcal{R}^d$ with respect to visual or textual tokens, where $d$ is the embedding length, we compute the language modeling loss (Equation 2) and propagate this loss backward through the Transformer blocks of the LLM. As shown in Figure 2a, we perform **Gradient Regularization** (GR) by identifying the $k$ largest and smallest gradient (Grad) values and setting them to 0, as follows.

$$i_{\max}, i_{\min} = \underset{k}{\arg\max}\, G, \underset{k}{\arg\min}\, G$$
$$G[i_{\max}] = G[i_{\min}] = 0 \tag{3}$$

This clipping will be performed on each token in both the Attention block and the MLP block of the Transformer blocks.

**Preserving low-level features.** LLMs typically consist of multiple stacked Transformer blocks, which enable the learned features to be specific to the model itself. Inspired by Deng et al. (2023), which suggests that preserving more low-level features in convolutional networks can improve cross-model transferability, we hypothesize that regularizing only the gradients of certain high-level features in LLMs can better balance the strength and transferability of attacks. Assuming that the LLM consists of $N$ Transformer blocks, we set $n = \lambda \cdot N$ and apply regularization only to the features of the last $n$ layers. The complete algorithm is presented in Algorithm 1.

**Relationships among various methods.** Our method builds upon previous work Moosavi-Dezfooli et al. (2017); Luo et al. (2024a); Zhang et al. (2023a); Wei et al. (2022), where correlations between them can be transformed by adjusting the values of hyperparameters, as illustrated in Figure 2b.

## 4 EXPERIMENTS

In this section, we validate the effectiveness of our method through extensive experiments and conduct an in-depth analysis of the factors influencing GrCPA.

Table 1: The results of targeted attacks against Flamingo. We investigate four types of tasks and report attack success rates for each task category. The last column represents the average performance across these four task types. To demonstrate the generalization of our method, we set multiple target answers. The best results are represented in **bold**.

| Target Answer | Method | VQA$_{general}$ | VQA$_{specific}$ | Classification | Captioning | Average |
|---|---|---|---|---|---|---|
| unknown | Single-P | 0.24 | 0.39 | 0.21 | 0.05 | 0.22 |
| | Multi-P | 0.67 | 0.86 | 0.64 | 0.31 | 0.62 |
| | CroPA | 0.92 | 0.98 | 0.70 | 0.34 | 0.74 |
| | GrCPA | **0.95** | **0.99** | **0.75** | **0.43** | **0.78** |
| I am sorry | Single-P | 0.21 | 0.43 | 0.47 | 0.34 | 0.36 |
| | Multi-P | 0.60 | 0.85 | 0.71 | 0.60 | 0.69 |
| | CroPA | 0.90 | 0.96 | 0.75 | 0.72 | 0.83 |
| | GrCPA | **0.94** | **0.96** | **0.83** | **0.82** | **0.88** |
| not sure | Single-P | 0.25 | 0.36 | 0.09 | 0.00 | 0.17 |
| | Multi-P | 0.55 | 0.55 | 0.11 | 0.02 | 0.31 |
| | CroPA | 0.88 | 0.86 | 0.30 | 0.17 | 0.55 |
| | GrCPA | **0.93** | **0.89** | **0.46** | **0.26** | **0.63** |
| very good | Single-P | 0.35 | 0.52 | 0.15 | 0.05 | 0.27 |
| | Multi-P | 0.81 | 0.93 | 0.40 | 0.20 | 0.59 |
| | CroPA | 0.95 | 0.97 | 0.64 | 0.27 | 0.71 |
| | GrCPA | **0.99** | **0.97** | **0.81** | **0.44** | **0.80** |
| too late | Single-P | 0.21 | 0.38 | 0.21 | 0.04 | 0.21 |
| | Multi-P | 0.78 | 0.90 | 0.54 | 0.17 | 0.60 |
| | CroPA | 0.90 | 0.95 | 0.73 | 0.20 | 0.70 |
| | GrCPA | **0.93** | **0.97** | **0.79** | **0.39** | **0.77** |
| metaphor | Single-P | 0.26 | 0.56 | 0.50 | 0.14 | 0.37 |
| | Multi-P | 0.83 | 0.92 | 0.81 | 0.42 | 0.75 |
| | CroPA | 0.96 | 0.99 | 0.92 | 0.62 | 0.87 |
| | GrCPA | **0.99** | **0.99** | **0.95** | **0.73** | **0.91** |

## 4.1 EXPERIMENTAL SETTINGS

**Datasets.** Given that VLMs tackle downstream tasks through visual question answering, it is imperative that the dataset encompasses both images and corresponding prompts. The images are sourced from the MS-COCO validation set (Lin et al., 2014). The VQA prompts are comprised of questions that are either general or specific to the image content, respectively referred to as VQA$_{general}$ and VQA$_{specific}$. The image-specific questions are derived from the VQA-v2 dataset (Goyal et al., 2017). Agnostic questions were constructed for VQA, with a focus on image classification and image captioning, ensuring a diverse range of lengths and semantic content.

**Models.** Without loss of generality, we evaluate the OpenFlamingo-9B (Alayrac et al., 2022; Awadalla et al., 2023), BLIP-2 (OPT-2.7B) (Li et al., 2023a; Zhang et al., 2022), LLaVA-1.5-7B(Liu et al., 2023) and InstructBLIP (Dai et al., 2023), which are influential models in the multimodal community.

**Parameters.** In alignment with previous research (Luo et al., 2024a), the image perturbation is configured to $16/255$, with $\alpha_1 = 1/255$, $\alpha_2 = 0.01$, and the number of iterations set to 1000. A maximum of 100 prompts are utilized for each individual sample. The proportion of Transformer blocks $\lambda$ is set to $1/4$; the update frequency $T$ is set to 1; and the number of extrema $k$ is also set to 1.

**Evaluation Metric.** In this paper, we report the Attack Success Rate (ASR) and facilitate the analysis by inducing the model to output specific text.

Table 2: Quantitative evaluation of attack stability. We assess the stability of different methods by determining whether the outputs at the 900th, 925th, 950th, 975th, and 1000th steps are consistent.

| Method | $VQA_{general}$ | $VQA_{specific}$ | Classification | Captioning | Average |
|--------|------------------|-------------------|----------------|------------|---------|
| Multi-P | 0.57 | 0.59 | 0.46 | 0.43 | 0.51 |
| CroPA | 0.61 | 0.65 | 0.53 | 0.49 | 0.57 |
| GrCPA | **0.67** | **0.71** | **0.57** | **0.53** | **0.62** |

## 4.2 COMPARISON WITH PREVIOUS METHODS

To comprehensively demonstrate the efficacy of our proposed GrCPA, we conducted a series of experiments using Flamingo (Awadalla et al., 2023), evaluating it against a range of target responses. The target text consists of statements such as `unknown`, `not sure`, and `I am sorry`, which indicate a deficiency in interpreting visual content, and `unknown` is the default setting for subsequent experiments unless otherwise specified. It also features phrases like `very good`, `too late`, and `metaphor`, which are irrelevant to the context.

Table 1 summarizes the evaluation results of targeted attacks. GrCPA outperforms previous SOTA methods across various experimental settings. Both the baseline methods and our method achieve higher attack success rates on the VQA task, likely due to the closer relationship between prompts and images in the VQA framework, where prompts more closely related to the image are more likely to enhance the effectiveness of the attack. This also indirectly demonstrates the sensitivity of adversarial samples to prompts in vision-language models, where adversarial samples may become ineffective when encountering different prompts. Furthermore, varying target answers can affect attack results. The experimental findings suggest that even rare and illogical responses, like metaphors, can still achieve high success rates. We also evaluate longer target answers, such as `I need a new phone`, in Appendix A.3. The results show that our method still outperforms the baseline methods. Additionally, we demonstrate the stability of our method through qualitative case studies in Appendix A.4.

To further validate the generalizability of our method, we also conduct experiments on LLaVA-1.5 and InstructBLIP, as detailed in Appendix A.5. These models are similarly susceptible to adversarial attacks, exhibiting serious security vulnerabilities. We also evaluate their cross-model transferability in Appendix A.7, but find weak transferability..

Besides showcasing the attack results, we also perform a quantitative analysis of the variations in attack stability across different methods. We further evaluate the stability of various attack methods by examining whether the model's outputs at the 900th, 925th, 950th, 975th, and 1000th iterations are consistent. As shown in Table 2, our method significantly enhances the stability of adversarial attacks across multiple tasks, which greatly aids in the large-scale evaluation of VLMs' robustness.

## 4.3 IMPACT OF PROMPT NUMBER

In this section, our focus is on examining the influence of the quantity of prompts utilized in the attack process on its effectiveness. We conduct attacks under various configurations, employing 1, 5, 10, 50, and 100 prompts against the BLIP-2 model with the objective of eliciting an `unknown` response.

As shown in Table 3, augmenting the quantity of prompts in the optimization phase substantially augments the cross-prompt transferability of visual adversarial samples. For example, escalating the number of prompts from one to ten results in a pronounced increment in the ASR of the baseline method, from 0.34 to 0.71, which corresponds to a more than twofold enhancement. The experimental outcomes clearly indicate that our methodology consistently outperforms the baseline approach across all configurations, thereby showcasing our method's superiority.

Nevertheless, augmenting the number of prompts directly leads to a substantial increase in the computational demands of adversarial attacks, presenting a significant impediment to the large-scale generation of visual adversarial samples. Additionally, there is a pronounced effect of diminishing marginal returns associated with increasing the number of prompts; beyond a threshold of 10 prompts,

Table 3: The results of the adversarial attack against BLIP-2. Different numbers of prompts are employed, and it is found that increasing the number of prompts improves the performance. The best performance values for each task are highlighted in **bold**.

| No. of Prompts | Method | $VQA_{general}$ | $VQA_{specific}$ | Classification | Captioning | Average |
|---|---|---|---|---|---|---|
| | Single-P | 0.24 | 0.34 | 0.45 | 0.32 | 0.34 |
| 1 | CroPA | 0.52 | 0.63 | 0.65 | 0.58 | 0.60 |
| | **GrCPA** | **0.55** | **0.65** | **0.69** | **0.60** | **0.62** |
| | Multi-P | 0.51 | 0.59 | 0.62 | 0.58 | 0.58 |
| 5 | CroPA | 0.81 | 0.83 | 0.80 | 0.84 | 0.82 |
| | **GrCPA** | **0.85** | **0.87** | **0.83** | **0.89** | **0.86** |
| | Multi-P | 0.68 | 0.81 | 0.68 | 0.67 | 0.71 |
| 10 | CroPA | 0.86 | 0.90 | 0.82 | 0.84 | 0.86 |
| | GrCPA | **0.88** | **0.93** | **0.84** | **0.85** | **0.87** |
| | Multi-P | 0.67 | 0.74 | 0.67 | 0.72 | 0.70 |
| 50 | CroPA | 0.90 | 0.93 | 0.87 | 0.91 | 0.90 |
| | **GrCPA** | **0.95** | **0.96** | **0.89** | **0.92** | **0.93** |
| | Multi-P | 0.67 | 0.76 | 0.68 | 0.66 | 0.69 |
| 100 | CroPA | 0.95 | 0.95 | 0.87 | 0.92 | 0.92 |
| | **GrCPA** | **0.99** | **0.99** | **0.93** | **0.95** | **0.96** |

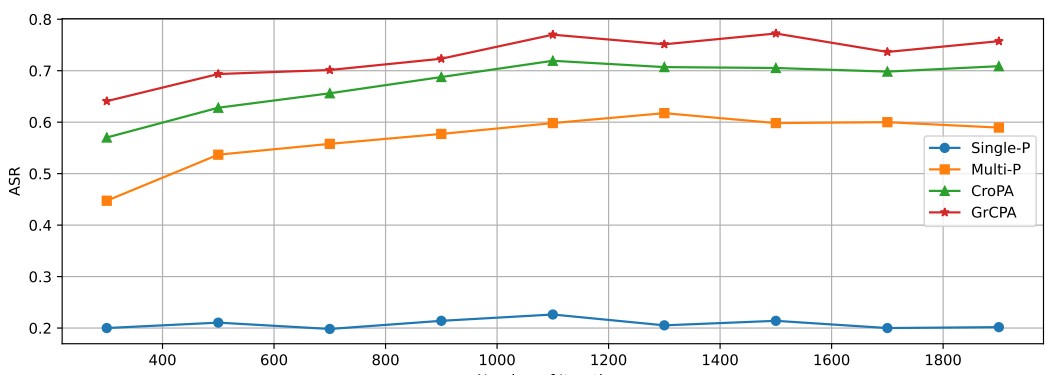

Figure 3: The impact of the number of iterations on attack success rate. Compared with the baseline algorithms, our method shows a certain degree of improvement across different numbers of iterations.

the enhancement in transferability becomes exceedingly constrained. Consequently, it is imperative to improve cross-prompt transferability using a finite set of prompts.

## 4.4 CONVERGENCE OF GRCPA

In this section, we explore the impact of the number of iterations during the optimization process on the attack success rate. All attacks are conducted using 10 prompts.

As shown in Figure 3, it can be observed that all methods show a corresponding increase in attack success rate with the number of iterations. This improvement is particularly evident in scenarios using multiple prompts, as more prompts necessitate learning more content.

Our method's performance gradually stabilizes after 1000 iterations. Compared to adversarial attacks on classification tasks, which typically require around 100 iterations, adversarial attacks on VLMs demand significantly more computational effort. However, our method can achieve better performance with fewer steps and demonstrates higher computational efficiency.

## 4.5 ABLATION STUDIES

In this section, we thoroughly analyze the effectiveness of GrCPA through ablation experiments.

Table 4: Ablation studies of gradient regularization.

| Method | VQA$_{general}$ | VQA$_{specific}$ | Classification | Captioning | Average |
|---|---|---|---|---|---|
| Single-P | 0.24 | 0.39 | 0.21 | 0.05 | 0.22 |
| Single-P(GR) | 0.29 | 0.45 | 0.24 | 0.11 | 0.27 |
| Multi-P | 0.67 | 0.86 | 0.64 | 0.31 | 0.62 |
| Multi-P(GR) | 0.77 | 0.91 | 0.71 | 0.35 | 0.78 |

Table 5: Ablation studies on single-modality regularization.

| Method | VQA$_{general}$ | VQA$_{specific}$ | Classification | Captioning | Average |
|---|---|---|---|---|---|
| GrCPA | 0.99 | 0.99 | 0.93 | 0.95 | 0.96 |
| GrCPA(Image) | 0.94 | 0.95 | 0.90 | 0.89 | 0.92 |
| GrCPA(text) | 0.92 | 0.93 | 0.86 | 0.87 | 0.89 |

Table 6: Ablation experiments on the impact of the layer proportion $\lambda$.

| $\lambda$ | 1 | 1/2 | 1/3 | 1/4 | 1/5 | 1/6 |
|---|---|---|---|---|---|---|
| ASR | 0.863 | 0.863 | 0.864 | 0.875 | 0.873 | 0.869 |

**The impact of gradient regularization.** Although GrCPAbuilds on previous work, this gradient regularization method is actually a general approach to reducing overfitting. As shown in Table 4, our experiments on Single-P and Multi-P demonstrate that it can provide cross-prompt transferability.

**The impact of regularizing different modalities.** In our method, we apply gradient regularization to both visual modality features and textual modality features in the LLM. In practice, it is feasible to regularize the feature gradients of a single modality. We conduct such experiments as shown in Table 5, but find that the attack success rate significantly decreased. Therefore, we believe that enhancing attacks on VLMs should consider both modalities whenever possible.

**The impact of proportion $\lambda$ of Transformer blocks.** We primarily test the effectiveness of gradient regularization on Transformer Blocks with LLMs at different proportions $\lambda$. The experimental results, as shown in Table 6, revealed that trimming only the last $1/4$ layers achieved the best performance. However, in terms of absolute performance, the differences among them were relatively minor.

## 5 CONCLUSION

In this paper, we investigate the adversarial robustness of large vision language models (VLMs). During our experiments, we first found that existing adversarial attacks on visual language models exhibit significant instability, with the optimization process for adversarial samples oscillating between success and failure. We believe that the root cause of this issue is overfitting during the optimization process, which poses a challenge to the large-scale generation of adversarial samples for visual language models. Furthermore, we experimentally investigated the effectiveness of adversarial attack enhancement methods that target only the visual modality within VLMs and found that these methods reduce attack performance. Based on these observations, we propose Gradient Regularized-based Cross-Prompt Attack (GrCPA), which clips the gradients of visual and textual features during error backpropagation, eliminating extreme gradients to prevent falling into local optima. Our regularization operation modifies only a small portion of the gradients and does not affect the convergence of the chain rule. Experiments on models such as BLIP-2 demonstrate that our method significantly improves the transferability of adversarial samples and confirms that current VLMs are sensitive to visual inputs and can be easily attacked. Therefore, we call on researchers to thoroughly evaluate the adversarial robustness of visual language models before deployment, especially in life-critical scenarios.

**Reproducibility.** In the experiments, we thoroughly report on the datasets, models, and parameter settings designed for this study, with all data being open-source and publicly available to ensure reproducibility.

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

# A  APPENDIX

## A.1  ATTEMPTS AT EMPLOYING UNIMODAL METHODS

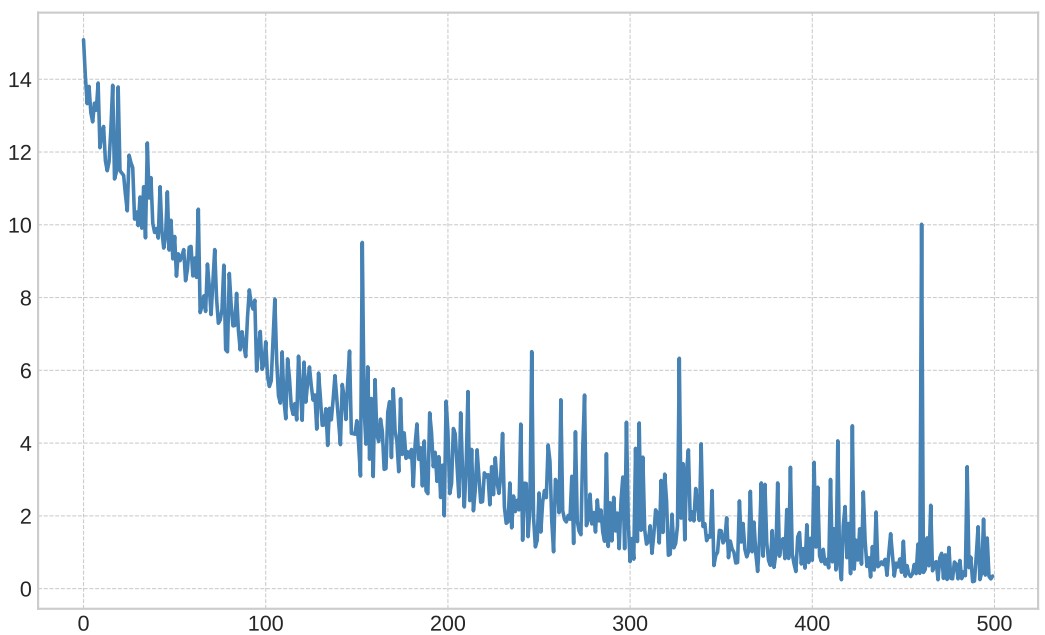

Figure 4: The impact of the number of iterations on attack success rate. Compared with the comparative algorithms, the method we propose shows a certain degree of improvement across different numbers of iterations.

In Figure 1, we employ a specific instance to elucidate the non-stationary nature of adversarial attacks on vision language models. We posit that the underlying cause of this non-stationary behavior is the overfitting that occurs during the optimization process. As depicted in Figure 4, we illustrate the variation in the loss during the optimization of adversarial samples. It can be observed that as the number of iterations increases, the loss diminishes progressively. However, the overall curve exhibits irregularities, particularly abrupt fluctuations, which may be attributed to the complexity of VLMs. Even minor alterations at the feature level can lead to significant variations in the output results.

Table 7: Comparison of different methods

| Method | Baseline | MI-FGSM | VMI-FGSM | DIM |
|--------|----------|---------|----------|-----|
| ASR | 0.71 | 0.69 | 0.65 | 0.43 |

To enhance the transferability of adversarial examples, a series of methods targeting visual models have been proposed, which we have attempted to apply to VLMs. Our primary experiments were conducted on MI-FGSM Dong et al. (2018), VMI-FGSM Wang & He (2021), and DIM Xie et al. (2019), with the first two methods focusing on enhancing adversarial attacks by correcting gradients to reduce overfitting, while the third approach emphasizes data augmentation techniques, such as padding and cropping. The experimental results are shown in Table 7.

Table 8: Evaluation of longer target answers.

| Target Answer | Method | VQA$_{general}$ | VQA$_{specific}$ | Classification | Captioning | Average |
|---|---|---|---|---|---|---|
| | Multi-P | 0.67 | 0.75 | 0.41 | 0.03 | 0.46 |
| I do not know | CroPA | 0.70 | 0.80 | 0.43 | 0.04 | 0.49 |
| | GrCPA | 0.73 | 0.81 | 0.55 | 0.11 | 0.55 |
| | Multi-P | 0.68 | 0.86 | 0.85 | 0.53 | 0.73 |
| I need buy a new phone | CroPA | 0.83 | 0.85 | 0.77 | 0.70 | 0.78 |
| | GrCPA | 0.85 | 0.86 | 0.78 | 0.73 | 0.80 |

## A.2 GrCPA Pipeline

---

**Algorithm 1** Gradient Regularization-based Cross-Prompt Attacks

---

**Require:** Model $f_\theta$, Target Text $a$, vision input $v$, prompt set $\mathcal{Q}$, perturbation size $\epsilon$, step size of perturbation updating $\alpha_1$ and $\alpha_2$, number of iteration steps $I$, adversarial prompt update interval $T$, number of LLM's Transformer blocks $N$, proportion of Transformer blocks $\lambda$

**Ensure:** Adversarial example $v'$

1: Initialise $v' = v$
2: **for** step =1 to $I$ **do**
3:     Uniformly sample the prompt $q_i$ from $\mathcal{Q}_M$
4:     **if** $q_i{'}$ is not initialised **then**
5:         Initialise $q_i' = q_i$
6:     **end if**
7:     Compute gradient for adversarial image : $g_v = \nabla_v \mathcal{L}(f_\theta(v', q_i), a)$:
8:         $g_v = \text{GR}(g_v)$
9:     Update with gradient descent: $v' = v' - \alpha_1 \cdot \text{sign}(g_v)$
10:     **if** mod(step, T) == 0 **then**
11:         Compute gradient for adversarial prompt: $g_q = \nabla_q \mathcal{L}(f_\theta(v', q_i), a)$:
12:         $g_q = \text{GR}(g_q)$
13:         Update with gradient ascent: $q_i' = q_i' + \alpha_2 \cdot \text{sign}(g_q)$
14:     **end if**
15:     Project $v'$ to be within the $\epsilon$-ball of $v$: $v' = \text{Clip}_{v,\epsilon}(v')$
16: **end for**
17: **return** $v'$

---

## A.3 Evaluation of Long Sequences

In the experimental results in Table 1, we report the effectiveness of attacks with varying word counts (e.g., 1 word, 2 words, 3 words). The results show that our method consistently produces effective attacks. To demonstrate that our method can handle different sequence lengths, we have included additional experiments with two other target sequences (e.g., `I do not know` and `I need a new phone`). Table 8 indicates that our attack method remains highly effective even with longer sequences. Of course, the effectiveness of the attack can vary significantly depending on the prompt for different tasks, which remains a promising direction for future research in cross-prompt studies.

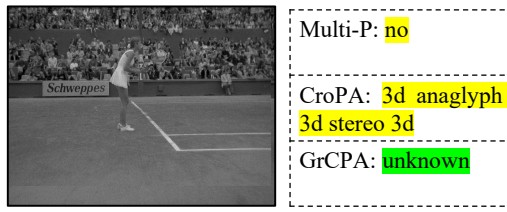

Figure 5: Qualitative evaluation of cross-prompt attacks on the BLIP-2 model.

## A.4 Qualitative Analysis

From the qualitative analysis of specific cases as shown in Figure 5, it can be observed that when encountering relatively blurry images, the success rate of all attacks decreases. The lack of high-frequency details in blurry images makes them less sensitive to the small local perturbations typically used in adversarial attacks. In such situations, CroPA can cause the model to produce nonsensical output (e.g., `3d anaglyph 3d stereo 3d stereo 3d stereo 3d stereo`), whereas our GrCPA does not cause the model to generate off-target outputs.

## A.5 Supplementary experiments on LLaVA and InstructBLIP

Considering that BLIP-2 and OpenFlamingo have been released for some time, we also conduct evaluations on the latest LLaVA-1.5-7B (Liu et al., 2023) and InstructBLIP-Vicuna-7b (Dai et al., 2023) to further validate the effectiveness of our method.

As shown in Table 9, the results on LLaVA 1.5 and InstructBLIP demonstrate that various attack methods can achieve high success rates, indicating a common weakness in vision-language models. Coupled with the experiments discussed in the main text, our proposed attack method proves to be highly effective across different architectures and parameter scales, further highlighting its generalization capability.

Table 9: Additional experiments on LLaVA and InstructBLIP.

| Method | LLaVA-1.5 | InstructBLIP |
|--------|-----------|--------------|
| Single-P | 0.34 | 0.31 |
| Multi-P | 0.89 | 0.90 |
| CroPA | 0.94 | 0.93 |
| GrCPA | 0.98 | 0.97 |

## A.6 Analysis of Regularization Methods

In our proposed method, we utilize a regularization technique by setting the gradient extremes to zero during backpropagation to reduce overfitting and enhance transferability, a strategy validated in some related works (Zhang et al., 2023a). However, we also thoroughly explore additional regularization techniques in this section, such as $L_2$ regularization.

Table 10: Additional experiments on more regularization methods.

| Method | VQA$_{general}$ | VQA$_{specific}$ | Classification | Captioning | Average |
|--------|-----------------|------------------|----------------|------------|---------|
| $L_2$ Regularization | 0.84 | 0.83 | 0.81 | 0.83 | 0.83 |
| GrCPA | 0.88 | 0.93 | 0.84 | 0.85 | 0.88 |

Table 10 summarizes our experimental results. $L_2$ regularization dot not appear to provide the desired performance improvement in our experiments, possibly due to its excessive influence on the overall gradient updates.

## A.7 Analysis of Cross-model Transferability.

We argue that there is a severe overfitting issue in vision-language models when faced with adversarial attacks, as they often require a high number of iterations, which makes the generated adversarial samples specific to their conditions (model and prompt). In Section 4, we provided a detailed

explanation of how GrCPA enhances cross-prompt transferability, and in this section, we discuss its impact on cross-model transferability.

Table 11: Evaluation of cross-model transferability from BLIP2-OPT2.7B to InstructBLIP-Vicuna-7B.

| Method | VQA$_{general}$ | VQA$_{specific}$ | Classification | Captioning | Average |
|--------|------------|-------------|----------------|------------|---------|
| Multi-P | 0.00 | 0.01 | 0.04 | 0.03 | 0.02 |
| CroPA | 0.00 | 0.04 | 0.15 | 0.11 | 0.07 |
| GrCPA | 0.01 | 0.05 | 0.19 | 0.11 | 0.09 |

We evaluate the transferability from BLIP2-OPT-2.7B to InstructBLIP-Vicuna-7B, as shown in Table 11. Both our method and the baselines exhibited weak adversarial transferability, likely because of the large architectural and parameter differences between these two models. Therefore, cross-model transferability warrants further investigation.

## A.8 ANALYSIS OF DEFENSES AGAINST GRCPA

Table 12: Evaluation of random rotation as a defense strategy.

| Method | VQA$_{general}$ | VQA$_{specific}$ | Classification | Captioning | Average |
|--------|------------|-------------|----------------|------------|---------|
| Multi-P | 0.58 | 0.79 | 0.52 | 0.26 | 0.53 |
| CroPA | 0.89 | 0.95 | 0.61 | 0.34 | 0.70 |
| GrCPA | 0.91 | 0.95 | 0.62 | 0.37 | 0.71 |

In this section, we use random rotations to initially investigate defense strategies as shown Table 12. It is evident that data preprocessing methods currently cannot effectively counter our adversarial attacks. In the future, we will continue to explore relevant adversarial training methods.

## A.9 PROMPTS FOR DIFFERENT TASKS

### A.9.1 PROMPTS FOR VQA

*Any cutlery items visible in the image?*
*Any bicycles visible in this image?*
*Any boats visible in the image?*
*Any bottles present in the image?*
*Are curtains noticeable in the image?*
*Are flags present in the image?*
*Are flowers present in the image?*
*Are fruits present in the image?*
*Are glasses discernible in the image?*
*Are hills visible in the image?*
*Are plates discernible in the image?*
*Are shoes visible in this image?*
*Are there any insects in the image?*
*Are there any ladders in the image?*
*Are there any man-made structures in the image?*
*Are there any signs or markings in the image?*
*Are there any street signs in the image?*
*Are there balloons in the image?*
*Are there bridges in the image?*
*Are there musical notes in the image?*
*Are there people sitting in the image?*
*Are there skyscrapers in the image?*

*Are there toys in the image?*
*Are toys present in this image?*
*Are umbrellas discernible in the image?*
*Are windows visible in the image?*
*Can birds be seen in this image?*
*Can stars be seen in this image?*
*Can we find any bags in this image?*
*Can you find a crowd in the image?*
*Can you find a hat in the image?*
*Can you find any musical instruments in this image?*
*Can you identify a clock in this image?*
*Can you identify a computer in this image?*
*Can you see a beach in the image?*
*Can you see a bus in the image?*
*Can you see a mailbox in the image?*
*Can you see a mountain in the image?*
*Can you see a staircase in the image?*
*Can you see a stove or oven in the image?*
*Can you see a sunset in the image?*
*Can you see any cups or mugs in the image?*
*Can you see any jewelry in the image?*
*Can you see shadows in the image?*
*Can you see the sky in the image?*
*Can you spot a candle in this image?*
*Can you spot a farm in this image?*
*Can you spot a pair of shoes in the image?*
*Can you spot a rug or carpet in the image?*
*Can you spot any dogs in the image?*
*Can you spot any snow in the image?*
*Do you notice a bicycle in the image?*
*Does a ball feature in this image?*
*Does a bridge appear in the image?*
*Does a cat appear in the image?*
*Does a fence appear in the image?*
*Does a fire feature in this image?*
*Does a mirror feature in this image?*
*Does a table feature in this image?*
*Does it appear to be nighttime in the image?*
*Does it look like an outdoor image?*
*Does it seem to be countryside in the image?*
*Does the image appear to be a cartoon or comic strip?*
*Does the image contain any books?*
*Does the image contain any electronic devices?*
*Does the image depict a road?*
*Does the image display a river?*
*Does the image display any towers?*
*Does the image feature any art pieces?*
*Does the image have a lamp?*
*Does the image have any pillows?*
*Does the image have any vehicles?*
*Does the image have furniture?*
*Does the image primarily display natural elements?*
*Does the image seem like it was taken during the day?*
*Does the image seem to be taken indoors?*
*Does the image show any airplanes?*
*Does the image show any benches?*
*Does the image show any landscapes?*
*Does the image show any movement?*
*Does the image show any sculptures?*

*Does the image show any signs?*
*Does the image show food?*
*Does the image showcase a building?*
*How many animals are present in the image?*
*How many bikes are present in the image?*
*How many birds are visible in the image?*
*How many buildings can be identified in the image?*
*How many cars can be seen in the image?*
*How many doors can you spot in the image?*
*How many flowers can be identified in the image?*
*How many trees feature in the image?*
*Is a chair noticeable in the image?*
*Is a computer visible in the image?*
*Is a forest noticeable in the image?*
*Is a painting visible in the image?*
*Is a path or trail visible in the image?*
*Is a phone discernible in the image?*
*Is a train noticeable in the image?*
*Is sand visible in the image?*
*Is the image displaying any clouds?*
*Is the image set in a city environment?*
*Is there a plant in the image?*
*Is there a source of light visible in the image?*
*Is there a television displayed in the image?*
*Is there grass in the image?*
*Is there text in the image?*
*Is water visible in the image, like a sea, lake, or river?*
*How many people are captured in the image?*
*How many windows can you count in the image?*
*How many animals, other than birds, are present?*
*How many statues or monuments stand prominently in the scene?*
*How many streetlights are visible?*
*How many items of clothing can you identify?*
*How many shoes can be seen in the image?*
*How many clouds appear in the sky?*
*How many pathways or trails are evident?*
*How many bridges can you spot?*
*How many boats are present, if it's a waterscape?*
*How many pieces of fruit can you identify?*
*How many hats are being worn by people?*
*How many different textures can you discern?*
*How many signs or billboards are visible?*
*How many musical instruments can be seen?*

### A.9.2    PROMPTS FOR IMAGE CLASSIFICATION

*Identify the primary theme of this image in one word.*
*How would you label this image with a single descriptor?*
*Determine the main category for this image.*
*Offer a one-word identifier for this picture.*
*If this image were a file on your computer, what would its name be?*
*Tag this image with its most relevant keyword.*
*Provide the primary classification for this photograph.*
*How would you succinctly categorize this image?*
*Offer the primary descriptor for the content of this image.*
*If this image were a product, what label would you place on its box?*
*Choose a single word that encapsulates the image's content.*
*How would you classify this image in a database?*

*In one word, describe the essence of this image.*
*Provide the most fitting category for this image.*
*What is the principal subject of this image?*
*If this image were in a store, which aisle would it belong to?*
*Provide a singular term that characterizes this picture.*
*How would you caption this image in a photo contest?*
*Select a label that fits the main theme of this image.*
*Offer the most appropriate tag for this image.*
*Which keyword best summarizes this image?*
*How would you title this image in an exhibition?*
*Provide a succinct identifier for the image's content.*
*Choose a word that best groups this image with others like it.*
*If this image were in a museum, how would it be labeled?*
*Assign a central theme to this image in one word.*
*Tag this photograph with its primary descriptor.*
*What is the overriding theme of this picture?*
*Provide a classification term for this image.*
*How would you sort this image in a collection?*
*Identify the main subject of this image concisely.*
*If this image were a magazine cover, what would its title be?*
*What term would you use to catalog this image?*
*Classify this picture with a singular term.*
*If this image were a chapter in a book, what would its title be?*
*Select the most fitting classification for this image.*
*Define the essence of this image in one word.*
*How would you label this image for easy retrieval?*
*Determine the core theme of this photograph.*
*In a word, encapsulate the main subject of this image.*
*If this image were an art piece, how would it be labeled in a gallery?*
*Provide the most concise descriptor for this picture.*
*How would you name this image in a photo archive?*
*Choose a word that defines the image's main content.*
*What would be the header for this image in a catalog?*
*Classify the primary essence of this picture.*
*What label would best fit this image in a slideshow?*
*Determine the dominant category for this photograph.*
*Offer the core descriptor for this image.*
*If this image were in a textbook, how would it be labeled in the index?*
*Select the keyword that best defines this image's theme.*
*Provide a classification label for this image.*
*If this image were a song title, what would it be?*
*Identify the main genre of this picture.*
*Assign the most apt category to this image.*
*Describe the overarching theme of this image in one word.*
*What descriptor would you use for this image in a portfolio?*
*Summarize the image's content with a single identifier.*
*Imagine you're explaining this image to someone over the phone. Please describe the image in one word?*
*Perform the image classification task on this image. Give the label in one word.*
*Imagine a child is trying to identify the image. What might they excitedly point to and name?*
*If this image were turned into a jigsaw puzzle, what would the box label say to describe the picture inside?*
*Classify the content of this image.*
*If you were to label this image, what label would you give?*
*What category best describes this image?*
*Describe the central subject of this image in a single word.*
*Provide a classification for the object depicted in this image.*
*If this image were in a photo album, what would its label be?*
*Categorize the content of the image.*

*If you were to sort this image into a category, which one would it be?*
*What keyword would you associate with this image?*
*Assign a relevant classification to this image.*
*If this image were in a gallery, under which section would it belong?*
*Describe the main theme of this image in one word.*
*Under which category would this image be cataloged in a library?*
*What classification tag fits this image the best?*
*Provide a one-word description of this image's content.*

### A.9.3 PROMPTS FOR IMAGE CAPTIONING

*Any cutlery items visible in the image?*
*Any bicycles visible in this image?*
*Any boats visible in the image?*
*Any bottles present in the image?*
*Are curtains noticeable in the image?*
*Are flags present in the image?*
*Are flowers present in the image?*
*Are fruits present in the image?*
*Are glasses discernible in the image?*
*Are hills visible in the image?*
*Are plates discernible in the image?*
*Are shoes visible in this image?*
*Are there any insects in the image?*
*Are there any ladders in the image?*
*Are there any man-made structures in the image?*
*Are there any signs or markings in the image?*
*Are there any street signs in the image?*
*Are there balloons in the image?*
*Are there bridges in the image?*
*Are there musical notes in the image?*
*Are there people sitting in the image?*
*Are there skyscrapers in the image?*
*Are there toys in the image?*
*Are toys present in this image?*
*Are umbrellas discernible in the image?*
*Are windows visible in the image?*
*Can birds be seen in this image?*
*Can stars be seen in this image?*
*Can we find any bags in this image?*
*Can you find a crowd in the image?*
*Can you find a hat in the image?*
*Can you find any musical instruments in this image?*
*Can you identify a clock in this image?*
*Can you identify a computer in this image?*
*Can you see a beach in the image?*
*Can you see a bus in the image?*
*Can you see a mailbox in the image?*
*Can you see a mountain in the image?*
*Can you see a staircase in the image?*
*Can you see a stove or oven in the image?*
*Can you see a sunset in the image?*
*Can you see any cups or mugs in the image?*
*Can you see any jewelry in the image?*
*Can you see shadows in the image?*
*Can you see the sky in the image?*
*Can you spot a candle in this image?*
*Can you spot a farm in this image?*

*Can you spot a pair of shoes in the image?*
*Can you spot a rug or carpet in the image?*
*Can you spot any dogs in the image?*
*Can you spot any snow in the image?*
*Do you notice a bicycle in the image?*
*Does a ball feature in this image?*
*Does a bridge appear in the image?*
*Does a cat appear in the image?*
*Does a fence appear in the image?*
*Does a fire feature in this image?*
*Does a mirror feature in this image?*
*Does a table feature in this image?*
*Does it appear to be nighttime in the image?*
*Does it look like an outdoor image?*
*Does it seem to be countryside in the image?*
*Does the image appear to be a cartoon or comic strip?*
*Does the image contain any books?*
*Does the image contain any electronic devices?*
*Does the image depict a road?*
*Does the image display a river?*
*Does the image display any towers?*
*Does the image feature any art pieces?*
*Does the image have a lamp?*
*Does the image have any pillows?*
*Does the image have any vehicles?*
*Does the image have furniture?*
*Does the image primarily display natural elements?*
*Does the image seem like it was taken during the day?*
*Does the image seem to be taken indoors?*
*Does the image show any airplanes?*
*Does the image show any benches?*
*Does the image show any landscapes?*
*Does the image show any movement?*
*Does the image show any sculptures?*
*Does the image show any signs?*
*Does the image show food?*
*Does the image showcase a building?*
*How many animals are present in the image?*
*How many bikes are present in the image?*
*How many birds are visible in the image?*
*How many buildings can be identified in the image?*
*How many cars can be seen in the image?*
*How many doors can you spot in the image?*
*How many flowers can be identified in the image?*
*How many trees feature in the image?*
*Is a chair noticeable in the image?*
*Is a computer visible in the image?*
*Is a forest noticeable in the image?*
*Is a painting visible in the image?*
*Is a path or trail visible in the image?*
*Is a phone discernible in the image?*
*Is a train noticeable in the image?*
*Is sand visible in the image?*
*Is the image displaying any clouds?*
*Is the image set in a city environment?*
*Is there a plant in the image?*
*Is there a source of light visible in the image?*
*Is there a television displayed in the image?*
*Is there grass in the image?*

*Is there text in the image?*
*Is water visible in the image, like a sea, lake, or river?*
*How many people are captured in the image?*
*How many windows can you count in the image?*
*How many animals, other than birds, are present?*
*How many statues or monuments stand prominently in the scene?*
*How many streetlights are visible?*
*How many items of clothing can you identify?*
*How many shoes can be seen in the image?*
*How many clouds appear in the sky?*
*How many pathways or trails are evident?*
*How many bridges can you spot?*
*How many boats are present, if it's a waterscape?*
*How many pieces of fruit can you identify?*
*How many hats are being worn by people?*
*How many different textures can you discern?*
*How many signs or billboards are visible?*
*How many musical instruments can be seen?*
*How many flags are present in the image?*
*How many mountains or hills can you identify?*
*How many books are visible, if any?*
*How many bodies of water, like ponds or pools, are in the scene?*
*How many shadows can you spot?*
*How many handheld devices, like phones, are present?*
*How many pieces of jewelry can be identified?*
*How many reflections, perhaps in mirrors or water, are evident?*
*How many pieces of artwork or sculptures can you see?*

