# OpenReview forum: "Gradient Regularization-based Cross-Prompt Attacks on Vision Language Models"
_ICLR.cc/2025/Conference — ICLR 2025 Conference Withdrawn Submission_

### Official Review · Reviewer_MTWv · 2024-10-31

**Soundness:** 3
**Presentation:** 3
**Contribution:** 2
**Rating:** 5
**Confidence:** 4

**Summary:**

The paper presents a novel Gradient Regularization-based Cross-Prompt Attack (GrCPA) targeting Vision-Language Models (VLMs), which addresses the issue of adversarial non-stationarity across diverse prompts. By leveraging gradient regularization, GrCPA mitigates the variability in adversarial success when multiple prompts are used. This approach enhances the robustness of adversarial examples, improving their transferability across prompts. Experiments on models such as Flamingo, BLIP-2, LLaVA, and InstructBLIP validate GrCPA’s effectiveness, showing superior attack stability and transferability compared to existing methods.

**Strengths:**

1. The method introduced is original. GrCPA’s use of gradient regularization for adversarial robustness across prompts introduces an effective method for enhancing VLM attack transferability.

2. The extensive experimental analysis across models and tasks (e.g., image captioning, VQA) confirms the soundness of the approach.

3. The method’s formulation and rationale are clearly articulated, supported by structured experiments that compare GrCPA with established baselines.

**Weaknesses:**

1. The technical depth of this paper is somewhat limited. Adversarial attacks are really not something that is surprisingly new in machine learning models, even in VLM. Incremental improvement in this area does not contribute much to this community. The method only introduces gradient normalization to stabilize the adversarial optimization, which is more like a trick for attack implementation.

2. I would expect some black-box transferability analysis to demonstrate the effectiveness of this attack.

**Questions:**

See the weakness section

---

### Official Review · Reviewer_eaZH · 2024-11-03

**Soundness:** 2
**Presentation:** 1
**Contribution:** 2
**Rating:** 1
**Confidence:** 4

**Summary:**

The similarities between this work and [1] in problem motivation, paper organization, experimental design, and writing text raise concerns about originality and potential plagiarism. Although the proposed approach does present some differences, the extent of overlap indicates that the authors may not have adequately distinguished their work from [1], published in ICLR 2024.

Here are some specific instances that suggest potential plagiarism:

1. The paper introduces a new problem, "cross-prompt transferability," which was first proposed in [1]. Notably, the main text, abstract, and introduction do not reference this prior work.

2. The organisation of this paper closely mirrors that of [1], with some tables being directly copied and merely modified to add an additional row.

3. Several paragraphs in this manuscript appear to be simple paraphrases of corresponding sections in [1].

4. In the experimental design, instead of acknowledging [1] as a basis, the authors claim to have independently designed the experiment, even though the design and details align precisely with those in [1].

There are additional similar issues present in the manuscript. Overall, this paper clearly does not adhere to accepted scientific writing standards.

[1] Luo, Haochen, et al. "An image is worth 1000 lies: Transferability of adversarial images across prompts on vision-language models." In The Twelfth International Conference on Learning Representations. 2023. Url: https://openreview.net/pdf?id=nc5GgFAvtk

**Strengths:**

N/A

**Weaknesses:**

N/A

**Questions:**

N/A

**Details Of Ethics Concerns:**

The similarities between this work and [1] in problem motivation, paper organization, experimental design, and writing text raise concerns about originality and potential plagiarism. Although the proposed approach does present some differences, the extent of overlap indicates that the authors may not have adequately distinguished their work from [1], published in ICLR 2024.

Here are some specific instances that suggest potential plagiarism:

1. The paper introduces a new problem, "cross-prompt transferability," which was first proposed in [1]. Notably, the main text, abstract, and introduction do not reference this prior work.

2. The organisation of this paper closely mirrors that of [1], with some tables being directly copied and merely modified to add an additional row.

3. Several paragraphs in this manuscript appear to be simple paraphrases of corresponding sections in [1].

4. In the experimental design, instead of acknowledging [1] as a basis, the authors claim to have independently designed the experiment, even though the design and details align precisely with those in [1].

There are additional similar issues present in the manuscript. Overall, this paper clearly does not adhere to accepted scientific writing standards.

[1] Luo, Haochen, et al. "An image is worth 1000 lies: Transferability of adversarial images across prompts on vision-language models." In The Twelfth International Conference on Learning Representations. 2023. Url: https://openreview.net/pdf?id=nc5GgFAvtk

---

### Official Review · Reviewer_8wzx · 2024-11-03

**Soundness:** 1
**Presentation:** 3
**Contribution:** 1
**Rating:** 3
**Confidence:** 4

**Summary:**

This paper addresses the challenge of creating transferable adversarial attacks across different prompts for vision language models (VLMs). The authors propose GrCPA (Gradient Regularized-based Cross-Prompt Attack), which utilizes gradient regularization to generate more robust adversarial examples.

**Strengths:**

1.	The experiments show the consistent better performance.
2.	The writing is easy to follow.

**Weaknesses:**

1.	The novelty and contribution are marginal. It only modifies the training loss in a very simple way.
2.	The logic is unconvincing to me. It is claimed that large gradients can lead to local optima and trigger overfitting issues. However, the Gradient Regularization simply sets the largest and the lowest gradients to zero. This raises two questions: (1) Why do you set the lowest gradient to zero? (2) Does the largest gradient represent a ‘large’ gradient? For example, in some cases, the largest gradient could be lower than the lowest gradient in another sample or batch. How do you define ‘large’ and ‘small’?"

**Questions:**

1.	During training and testing, do you use the same text prompts? If so, it seems that cross-prompt is just overfit on several prompts rather than one.

---

### Official Review · Reviewer_MwLh · 2024-11-04

**Soundness:** 2
**Presentation:** 2
**Contribution:** 1
**Rating:** 1
**Confidence:** 4

**Summary:**

The authors proposed a method termed Gradient Regularized-based Cross-Prompt Attack (GrCPA) that creates adversarial images that transfer across prompts. The GrCPA method extends the previous cross-prompt framework by applying gradient regularisation. The effectiveness of the GrCPA is evaluated with Flamingo, BLIP-2, LLaVA, and InstructBLIP on different tasks.

**Strengths:**

- The paper conducts extensive experiments on various VLMs to prove the effectiveness of GrCPA.
- The paper is easy to follow.

**Weaknesses:**

- **The novelty is limited**: As detailed in Section A.2 (line 878), the only difference between GrCPA and a recent work termed CroPA [1] is the addition of Gradient Regularization. The pipeline of GrCPA is highly similar to that of CroPA.

- **Practical applicability to the real world is limited**: As shown in Table 11, GrCPA does not demonstrate strong transferability across different models, with the average ASR remaining below 10%.

[1] Luo, Haochen, Jindong Gu, Fengyuan Liu, and Philip Torr. "An image is worth 1000 lies: Transferability of adversarial images across prompts on vision-language models." In The Twelfth International Conference on Learning Representations. 2023.

**Questions:**

- What is the impact of using different extrema K?

- Why the top k largest values of the gradient vectors are clipped directly to 0 instead of other constant values?

**Details Of Ethics Concerns:**

The overall structure of the paper closely resembles that of a recent work [1]. Notably, the experimental section fails to adequately acknowledge or reference the results and insights (such as the results on high ASR when targeted text is set to nonsensical phrases) in [1]. This omission creates the impression that these experiments are entirely original to the authors.

[1] Luo, Haochen, Jindong Gu, Fengyuan Liu, and Philip Torr. "An image is worth 1000 lies: Transferability of adversarial images across prompts on vision-language models." In The Twelfth International Conference on Learning Representations. 2023.

---

### Note · Authors · 2024-11-28

I have read and agree with the venue's withdrawal policy on behalf of myself and my co-authors.